# The potential virulence of *Listeria monocytogenes* strains isolated from fresh produce processing facilities as determined by an invertebrate *Galleria mellonella* model

Umaru Bah[1], Rosa de Llanos Frutos[2], Samantha Donnellan[1], Alva Smith[1], Allen Flockhart[1], Ian Singleton[1], Nick Wheelhouse[1] *

1 School of Applied Sciences, Edinburgh Napier University, Edinburgh, Scotland, United Kingdom,
2 Universitat Jaume I, Castell de la Plana, Valencia, Spain

* n.wheelhouse@napier.ac.uk

**Data Availability Statement:** All relevant data are within the manuscript and its Supporting Information files.

## Abstract

*Listeria monocytogenes*, a bacterium responsible for listeriosis, is an environmental and food-borne pathogen that poses a particular risk to pregnant women and the elderly. While traditionally associated with animal products, ready-to-eat salads are increasingly recognised as a source of Listeria outbreaks. However, little is known about the potential virulence of Listeria isolates from the fresh produce environment. This study assessed the virulence potential of nine *L. monocytogenes* strains from the fresh produce chain using the *Galleria mellonella* invertebrate infection model. Larvae were infected with $10^6$ CFU of each strain via their circulatory system and compared to a reference strain *L. monocytogenes* (EGD-e) and *Listeria ivanovii*. Virulence was evaluated by measuring mortality rates, health index score of larvae, viable bacterial counts in the larvae, and the larvae's immune. Significant differences in larval mortality were observed among strains. Strains NLmo4 and NLmo5 caused the highest mortality rates (98.8% and 96.7%, respectively at 7 days post-infection), while strain NLmo20 had a significantly lower mortality rate of 65% at the same time point (p<0.05). Six isolates that caused varied mortality rates were then selected and tested for their ability to replicate both *in vitro* and *in vivo* and their impact on larval haemocyte density. *In vitro* growth rates were not significantly different among *L. monocytogenes* strains or compared to *Listeria ivanovii*. However, *L. monocytogenes* strains persisted and replicated in larvae up to 7d days post-infection, whereas *Listeria ivanovii* was reduced by 5 logs CFU by day 7. The presence of these *L. monocytogenes* strains caused organ damage in larvae, indicated by increased melanisation and subsequent larval death. Haemocyte density showed insignificant fluctuations following infection. In conclusion, the results of this study suggest *L. monocytogenes* strains from fresh produce food chain have varying pathogenicity levels and can pose potential risk to human health.

**Funding:** This study was financially supported by Edinburgh Napier University in the form of a Research Excellence Grant received by NW, IS, and RL. No additional external funding was received for this study.

**Competing interests:** he authors have declared that no competing interests exist.

## Introduction

*Listeria monocytogenes* (*L. monocytogenes*) is a Gram-positive, rod-shaped, facultative intracellular anaerobic bacteria. The organism is the causative agent of listeriosis, with the elderly, immunosuppressed, pregnant women and new-born infants particularly at risk [1, 2]. Although incidence of the disease is low compared to other foodborne pathogens [3], the clinical outcome is often more serious. In mild cases the illness will lead to febrile gastroenteritis, however, can result in septicaemia, meningitis, and prenatal complications in expectant mothers (preterm birth, miscarriage, stillbirth) [4]. Survivors often end up with lasting neurological conditions (cognitive impairment, hearing loss, focal neurological deficits, and epilepsy) [5].

In the European Union and European Economic Area (EU/EEA) *L. monocytogenes* infections are relatively rare with an incidence rate of 0.62 per 100,000 [6]. Nevertheless, the number of reported listeriosis cases is steadily rising as 2,772 cases were reported in 2022, a 58.04% rise from total reported cases in 2012 [7]. Fatality rates from *L. monocytogenes* infections are also tremendously high (20–30%) relative to other bacterial and fungal infections ($\leq$1%) making the bacterium an increasing public health concern [6]. In the USA, for instance, though *L. monocytogenes* is estimated to cause 1,591 foodborne ill health cases annually (0.02% of total cases) it accounts for 18.9% of all foodborne related deaths. By comparison, *Salmonella* accounts for 646 times as many cases but only causes 123 more deaths (0.5% mortality rate), and although *E. coli* causes 40 times as many cases it causes 92.2% less deaths annually as compared to *L. monocytogenes* [8]. These make listeriosis a notifiable disease of great concern and reporting it as mandatory in all EU member states, including the United Kingdom (EU Directive 2003/99/EC).

The bacterium can contaminate fresh produce, processed ready-to-eat (RTE) meats, and dairy products [9–11]. Currently, all *L. monocytogenes* strains are treated the same for regulatory purposes, thus the presence of *L. monocytogenes* on foodstuffs is sufficient to result in product recall and this costs the UK economy an estimated £245M annually [12]. However, *L. monocytogenes* is genetically diverse and *in silico* predictions suggest there are significant differences in the disease-causing potential of different strains of the bacterium. Therefore, it is important to determine the virulence potential of isolates of the bacterium in order to evaluate the human health risks they pose.

In a previous study we characterised, using molecular typing, a number of genetically diverse *L. monocytogenes* strains derived from RTE produce and its processing environment which differed in their carriage of virulence factors and *in vitro* biofilm forming potential [13]. This current study was carried out to determine the virulence of a number of those previously characterised strains using the *Galleria mellonella* (*G. mellonella*) invertebrate infection model.

## Materials & methods

### Bacterial isolates used in this study

The virulence of ten *L. monocytogenes* strains were evaluated in this investigation. All strains, except the EGD-e (reference strain, NCTC7973), were obtained from a commercial food testing laboratory. These strains were isolated from various stages of the fresh produce supply chain (FPSC) around the UK between May 2016 and April 2017. Strains were isolated using ISO11290-2: 2017, species identification carried out using biochemical tests (API Listeria, bioMerieux/Microbact Listeria, Thermo Scientific) and strain identification by whole genome sequencing in an earlier publication [13].

*L. monocytogenes* is a model organism for investigating host-pathogen interactions, resulting in significant advancements across various disciplines [14], and EGD-e is the most

characterised *L. monocytogenes* strain [15], which has resulted in its inclusion in this study as a reference strain. Also, to ensure that potential differences in virulence observed are not species dependent a non-*L. monocytogenes* strain, *Listeria ivanovii* (NCTC12701), known for its rarity in causing human infections, was used as a second reference strain. This strain was also isolated from a final product spinach sample and identified by whole genome sequencing in the same aforementioned study.

The strains were processed for long-term storage at Edinburgh Napier University (ENU) as described by Smith *et al.* [13], suspended in BHI broth and 50% glycerol, and frozen at -80°C until required for virulence testing.

## Bacterial growth rate and inoculum size determination

The *L. monocytogenes* strain EGD-e (reference strain, NCTC7973) was grown overnight in a shaken incubator (200rpm, 37°C) in broth culture of BHI (BHI, Oxoid) media. The optical density ($OD_{600nm}$) was recorded and the cultures diluted to a starting $OD_{600nm}$ of 0.05, incubated under the same experimental conditions for 7 h and $OD_{600nm}$ recorded at hourly intervals. At hourly intervals viable bacterial colony forming units (CFU) counts were also enumerated by serially diluting and plating out cultures on Oxford agar base (Oxoid Ltd) supplemented with Listeria selective antibiotics amphotericin B (10μg/ml), colistin sulphate (20μl/ml), acriflavin (5μl/ml), ceotetan (2μl/ml), and fosfomycin (10μl/ml). Spread plates were incubated at 37°C for 24 h and CFU were counted to generate a calibration curve of $OD_{600nm}$ and Log CFU.

## Preparation of bacterial cultures for *G. mellonella* infections

To prepare bacterial inoculum for *G. mellonella* larvae infection, 10ml BHI broth was inoculated with a single CFU from a pure subculture and incubated overnight (37°C, 200rpm, aerobic conditions). Cultures were diluted in BHI broth to an $OD_{600nm}$ 0.42, which equated to $10^9$ CFU $ml^{-1}$. Cells were harvested by washing twice in PBS (4200 rpm, 10 min, 22°C), and diluted to a required dose for larvae infection.

## Infection of *G. mellonella* larvae and monitoring

*G. mellonella* larvae were purchased from UK Waxworms *Ltd* (Sheffield, UK). Larvae were stored at 20°C and used for bacterial challenge within 24 h of delivery to the lab. Only healthy-looking larvae weighing between 0.25–0.35g with no signs of melanisation were used, and a new batch of larvae was used in each experimental replicate. *G. mellonella* larvae (n = 30) were injected with $10^6$ CFU/larva delivered in 20μl PBS through the last right proleg using an insulin syringe, as described previously [16]. As control, larvae (n = 30) were inoculated with 20μl PBS. Bacterial inoculum dose for each set of experiments was confirmed by serially diluting and platting out inoculum dose on Oxford agar base (Oxoid Ltd).

## *Galleria mellonella* larvae monitoring post infection and health index scoring system (HISS)

*G. mellonella* larvae were examined individually on daily basis post infection. Larvae mortality was assessed by turning larvae on their backs and checking for leg movement; healthy larvae upright themselves while the dead showed no movement. To measure more subtle differences in larvae health status the health index scoring system (HISS), earlier described by Loh et al. [17] was adapted. The HISS is useful in determining subtle and otherwise unnoticeable differences in virulence if using only the binary assessments of dead or alive [18, 19]. It enables

**Fig 1. *G. mellonella* health index scoring system (HISS).** (A) Scores ascribed to different larvae attributes post-inoculation. (B) Larvae melanisation: uninfected larva showing no melanisation (1), infected larva with <3 melanisation spots (2), larva with >3 melanisation spots (3), brown larva (4), black and typically dead larva (5). (C) Pupae (fully formed cocoon) with arrow indicating silk formation. Table (A) is as adapted from [17].

greater reproducibility and comparison of data between laboratories, and as a result is gaining wider usability [20–22]. We therefore applied the HISS by examining each larvae for the following attributes: larvae activity, cocoon formation, cuticle melanisation, and survival (Fig 1). Individual larva scores were then aggregated to give a final score for each inoculum dose.

### *In vitro* growth rate of *L. monocytogenes* strains

Three *L. monocytogenes* strains (NLmo4, NLmo14, and NLmo20) were selected as representative strains based on their pathogenicity in larvae for further investigation. NLmo4 induced the highest larvae mortality and NLmo14 induced intermediate mortality relative to other strains tested whilst NLmo20 caused least larvae deaths at 7 d post infection. Growth rate of these strains *in vitro* was assessed and compared to the *L. monocytogenes* reference strain EGD-e, and *L. ivanovii*.

Overnight planktonic cultures were diluted to a starting $OD_{600nm}$ of 0.05 in 50ml BHI broth. At hourly intervals $OD_{600nm}$ was determined and CFU counts enumerated at bi-hourly intervals over a 12 h time course. Calibration curves of Absorbance and Log CFU were generated to determine bacterial growth over time.

### Determination of Listeria load in *G. mellonella* larvae

*L. monocytogenes* bacterial burden was evaluated as previously described [23]. Briefly, larvae (75 larvae per treatment) were inoculated with $10^6$ CFU/larva. At fixed time points, over a 7d period, three live larvae were homogenised using a Stomacher (Stomacher® 80 Biomaster, Seward, UK) in 3ml of sterile PBS. Homogenate was serially diluted in PBS and aliquots of 100µl plated on Oxford Listeria plates containing amphotericin B (10µg/ml), colistin sulphate (20µl/ml), acriflavin (5µl/ml), ceotetan (2µl/ml), and fosfomycin (10µl/ml) to inhibit growth of native larval flora and allow Listeria selection. Plates were incubated at 37˚C for 48 h and Listeria CFU per larvae enumerated. Experiments were performed independently three times using different larvae batch each time and means ± SD determined.

### Determination of Larval haemocyte density post Listeria inoculation

Larvae were inoculated with $10^6$ CFU and the haemocyte density was assessed daily for 7 d. At each time point post-infection three alive larvae were pierced at the side of the head with a

sterile needle and the haemolymph pooled together into a pre-chilled Eppendorf containing phenylthiourea granules to prevent melanisation, as carried out previously [24]. Haemolymph was diluted in PBS containing 0.37% (v/v) 2-Mercaptoethanol and cell density assessed using a haemocytometer. Three independent experiments were performed and the means ± SEM were expressed as haemocytes per ml of haemolymph.

### Data analysis

We used GraphPad Prism v 10.0.2 (GraphPad Software, San Diego, CA, USA) for statistical analysis. Survival data were plotted using the Kaplan-Meier method or survival heat-map, and comparisons between treatment groups were made using the log-rank Mantel-Cox test. Statistical tests for significance on Health Index Scores, and Larvae haemocytes density were performed using Two-Way ANOVA with $p \leq 0.05$ considered significant. All experiments were performed a minimum of three times.

## Results

### Larvae health and survival with *EGD-e*

The overall health index of larvae (Fig 2(A)) was found to be dose-dependent. As early as 24 h post-infection $10^7$ CFU induced significant differences in larval health in comparison with the non-infected control and lowest dose ($10^5$ CFU; $p < 0.01$). The health scores declined rapidly post this time-point to a score of zero at 4 days post-infection for the dose of $10^7$ CFU. There were no significant differences in health scores observed between PBS-inoculated larvae (control) as compared with the lowest bacterial, $10^5$ CFU, throughout the course of the experiment ($p = 0.642$). No significant melanisation was observed in PBS-inoculated control larvae, whilst all significant melanisation observed within the $10^5$ CFU dose resulted in larval death. The $10^6$ CFU treatment showed a steady decline in larval health correlating with that observed for larval survival for the same dose. Using Kaplan-Meier survival curves (Fig 2(B)) a dose-dependent survival was observed, similar to that obtained using the health index score data. The dose of $10^7$ CFU had an $LD_{50}$ of under 3 d, and 4 d for the dose of $10^6$ CFU. Larval survival using $10^5$ CFU (sub-lethal dose) was also similar to that observed in PBS-inoculated control larvae during the 7 d course of experimentation.

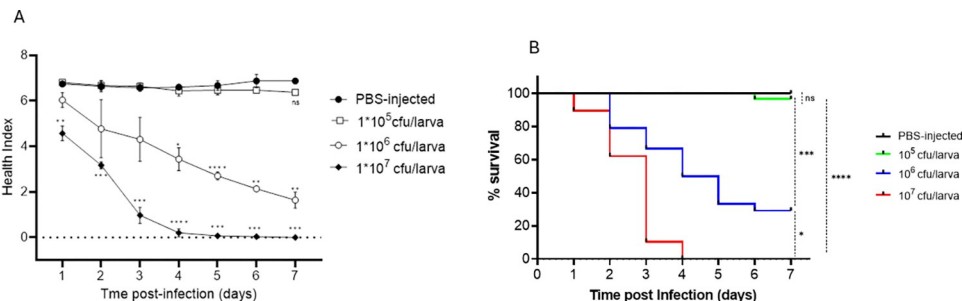

**Fig 2. *L. monocytogenes* strain EGD-e infection of *G. mellonella* induces dose-dependent reduction in health scores and survival.** (A) Larvae health were scored base on activity, cocoon formation, melanisation, and survival (B) Kaplan-Meier survival curves of *G. mellonella* larvae post-inoculation with EGD-e. Bacterial cultures were grown in BHI to stationary phase, washed twice and re-suspended in PBS. Larvae were inoculated with one of three doses, controls with 20µl PBS and subsequently incubated at 37˚C. All three doses caused time-dependent mortality of larvae with $10^7$ CFU inducing the highest mortality. $10^5$ CFU caused insignificant larval mortality and $10^6$ CFU induced gradual larval death. Results are Mean of three independent tests. Statistical comparisons are survival rates at day 7 post-infection (* $p < 0.05$; *** $P<0.001$; **** $P<0.0001$; ns, no significant differences).

### *Listeria monocytogenes* virulence in *G. mellonella* is strain and time dependent

PBS-inoculated larvae remained alive during the 7 d course of this investigation with similar observations also made for *L. ivanovii* inoculated larvae (96 ± 4.4%) survival at 7 d post-infection (Fig 3(A)). *L. monocytogenes* inoculated larvae showed increasing mortality rates over time indicating cumulative bacterial pathogenesis, which was analogous to earlier reports using clinical and mutant isolates of *L. monocytogenes* in *G. mellonella* [25]. Inter-strain comparisons revealed strain specific variations in virulence based on the mortality rates and overall larvae survival at the end of the 7 d time course. Inoculation with different Listeria strains caused significant differences in *G. mellonella* larval mortality (F = 23.69, df = 11, 66, p < 0.0001). Notably, NLmo4, NLmo5 & NLmo7 induced >90% mortality at 7 d post-infection while a 4.4 ± 4.4% mortality was observed in *L. ivanovii* inoculated larvae at the same time point.

All *L. monocytogenes* strains induced significantly higher larvae mortality than *L. ivanovii* (p < 0.05), and all, except NLmo20, were also found to be more virulent in *G. mellonella* than the reference strain (*EGD-e*). Amongst the investigated *L. monocytogenes* strains NLmo4 and NLmo5 induced the highest mortality rates (98.8 ± 1.1% and 96.7 ± 1.9%, respectively) and NLmo20 caused the least larval deaths (65±2.9%), an over 30 percentage points variation indicating significant differences in strains virulence. Additionally, varied mortality rates were observed among *L. monocytogenes* lineage I strains, albeit were statistically insignificant (p ≥ 0.05). When compared to lineage II isolates statistical significance was only observed between NLmo6 (lineage I) and NLmo4 and NLmo5 (both lineage II).

As a collective, *L. monocytogenes* lineage types did not correlate with virulence (Fig 4). Lineage II strains (three strains) induced higher mortality in larvae as compared with lineage I strains (six strains), however, mean differences between the two lineages were not significant (p = 0.567, Log-rant test). Similar observations were made for the mean health index scores for the two groups.

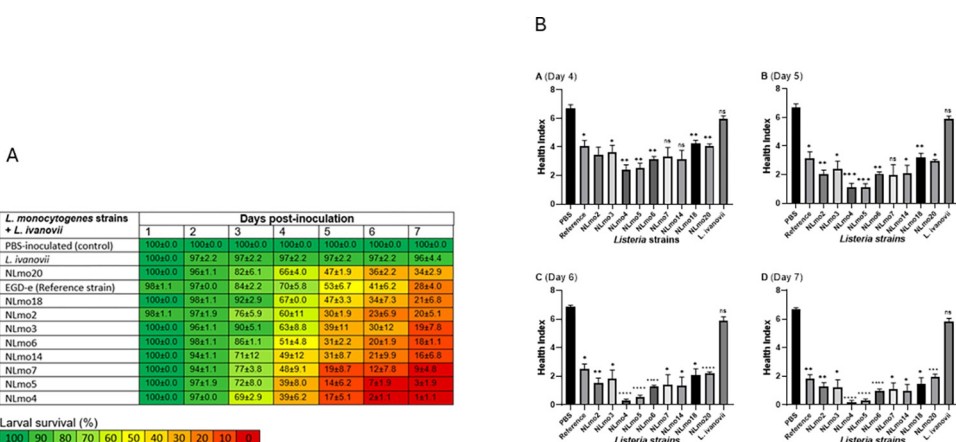

**Fig 3. Bacterial strain, and time-dependent, effect of infection on *G. mellonella* larvae survival and health status.** Larvae were inoculated with a stationary phase dose of 10^6 CFU for each of the tested strains, and controls with 20μl of PBS and subsequently incubated at 37˚C. (A) Larval mortality was monitored daily and was assessed by turning larvae on their backs and checking for leg movement. (B) Larvae health were scored base on activity, cocoon formation, melanisation, and survival. Results are Mean ± SD of three independent tests. (A-D) overall larvae health status at 4-, 5-, 6-, and 7 d post infection, respectively. Statistical differences are as compared to PBS-inoculated larvae (control) *p< 0.05; **p< 0.01; *** P<0.001; **** P<0.0001; ns, no significant differences (Two-Way ANOVA).

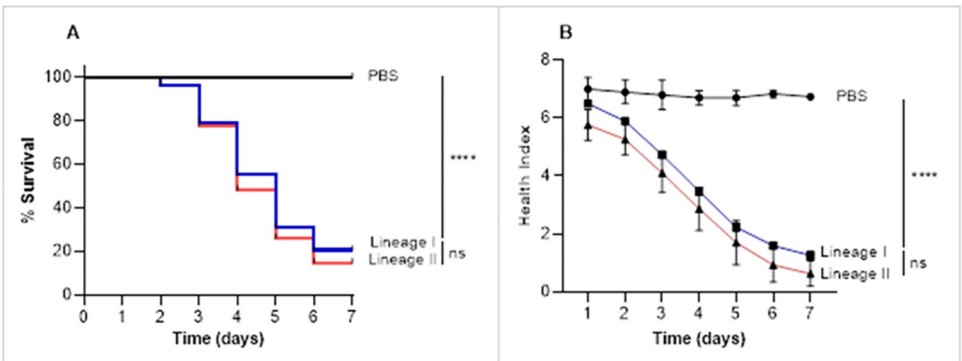

**Fig 4. Virulence of *L. monocytogenes* lineages in *G. mellonella*.** Virulence of six *L. monocytogenes* lineage I and four lineage II strains were assessed. Larvae were inoculated with a dose of $10^6$ CFU/larvae of stationary phase cultures and monitored daily for 10d. (A) Kaplan-Meier survival curves. ****$p < 0.0001$, ns, not significant (Log-rank test). (B) Mean ± SEM health index scores of larvae. **** $p< 0.001$, ns not significant (Two-Way ANOVA). Results represent three independent experiments.

### Listeria isolates have similar growth kinetics *in vitro*

Given the earlier determined inter strain differences in virulence in *G. mellonella* larvae, isolates representative of the least to the potentially most virulent strains were selected for further assessment. Bacterial growth rate *in vitro* can be used as a predictor of strain virulence in hosts. To establish whether *L. monocytogenes* strains used in this study have different growth capabilities in laboratory media, mean growth rates in BHI broth was determined over a 12 h time course as described.

Starter cultures, at $OD_{600nm}$ of 0.01, contained mean CFU of $10^7$ per ml (S1 Fig). Rapid growth was observed amongst all *L. monocytogenes* strains within the first 6 h (exponential growth), consistent with reported growth rates for *L. monocytogenes* EGD-e in BHI media at 37˚C [26] Over the same time points *L. ivanovii* exhibited slower growth relative to *L. monocytogenes* strains, as compared to EGD-e a difference of 3.08 x$10^9$ CFU ml$^{-1}$ was observed. Intra strain comparisons showed no statistical significance in growth rates amongst the *L. monocytogenes* strains within the first 6h ($p \geq 0.05$), except between EGD-e and NLmo14 at 6h ($p = 0.0311$). In contrast, significant differences in viable CFU counts were observed when *L. monocytogenes* strains were compared to *L. ivanovii* at both 4 h and 6 h with differences been more significant at 6h especially between EGD-e and NLmo14 when compared to *L. ivanovii* (S1 Fig, $p < 0.0001$, Two-Way ANOVA). At ~7 h all isolates, except *L. ivanovii*, reached stationary growth as observed in both CFU ml$^{-1}$ and $OD_{600nm}$ measurements. Though *L. ivanovii* had a slower growth rate, after 9 h incubation it was observed to have CFU counts comparable to that observed in the other isolates ($p \geq 0.05$, Two-Way ANOVA). During the 12 h time course bacterial growth increased by ~2 Logs for all isolates plateauing at $10^9$ CFU ml$^{-1}$. Though NLmo14 and NLmo20 attained the highest mean CFU ml$^{-1}$ no significant differences were observed amongst all the strains investigated at the stationary phase as all isolates reached the same CFU ml$^{-1}$ and final $OD_{600nm}$ by 12 h. Therefore, the bacterial replication *in vitro* did not correlate with the prior observed differences in virulence by number of larval mortalities caused.

### *L. monocytogenes* infection of *G. mellonella* larvae was accompanied by bacterial growth

Viable CFU counts decreased for the first 6 h post-infection for all isolates with replication rates varying thereafter (Fig 5). However, recovered bacterial counts from larvae in the first 6 h

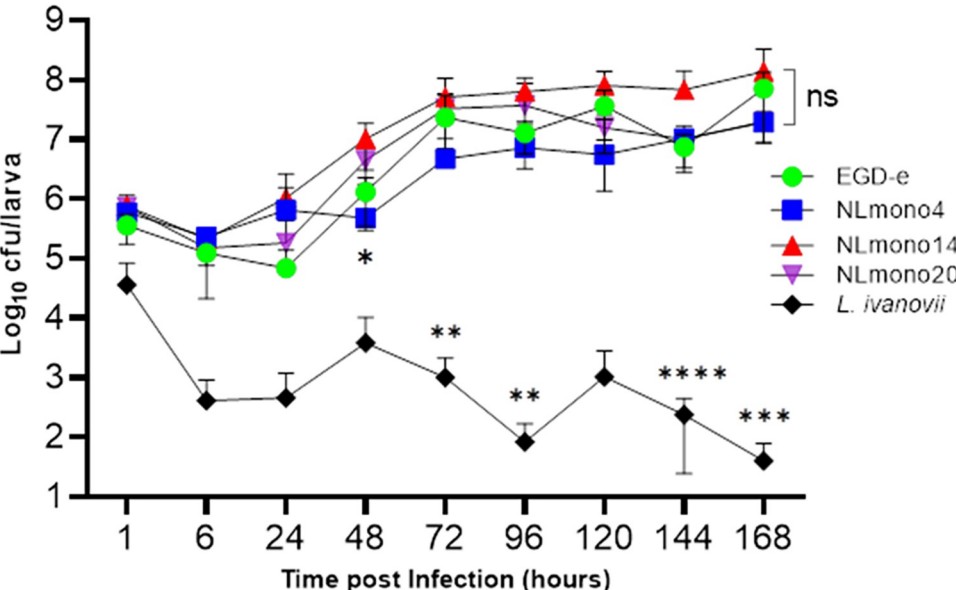

**Fig 5. Recovery of *L. monocytogenes* strains EGD-e, NLmo4, NLmo14, and NLmo20, and *L. ivanovii* (absorbance and CFU counts) as a function of time.** BHI broth was seeded with bacterial cultures in stationary phase to a starting absorbance (600 nm) of 0.01 and incubated at 37°C (200 rpm, aerobic conditions). Absorbance was measured hourly and CFU ml$^{-1}$ at bi-hourly intervals. Results represent individual replicates (CFU ml$^{-1}$) and mean ± SEM values of three independent determinations.

post-infection did not significantly differ from the dose used to initiate infection for all strains, except *L. ivanovii* (p = 0.001 and p < 0.0001, at 1 h and 6 h post-infection, respectively). After 24h bacterial burden was observed to increase rapidly in all L. monocytogenes infected larvae and plateaued after 72 h. This corresponded to ~2 Logs increase in bacterial burden from the infecting bacterial dose. Though NLmo14 consistently exhibited higher replication rates in vitro final bacterial burden in *G. mellonella* did not significantly differ amongst *L. monocytogenes* strains at the end of the 7 d time course (p = 0.488, one-Way ANOVA). In contrast, bacterial burden significantly differed between *L. monocytogenes* and *L. ivanovii* inoculated larvae. Viable *L. ivanovii* CFU counts gradually declined during the 7 d time course decreasing by a total of 5 Logs at 7 d post-infection. This strain was unable to establish an infection indicating it is relatively avirulent at a dose of 10$^{6}$ CFU in *G. mellonella* larvae, and these results corresponded with its relatively low mortality rates. *G. mellonella* was therefore effective at clearing the *L. ivanovii* infection, which was demonstrated by a lack of recoverable CFU even in neat samples at specific time points in the investigated strains, except *L. ivanovii*.

## Inoculation of *G. mellonella* larvae with *Listeria* leads to alterations in haemocyte density

Haemocytes are the main cellular component in *G. mellonella* immunity and they mediate the larvae immune response to infections [27]. Fluctuations in *G. mellonella* haemocyte densities following exposure to a range of microorganisms have been demonstrated [24, 28, 29]. These fluctuations, in addition to microbial load, have been suggested as indicators of microbial pathogenicity in *G. mellonella* larvae [30]. Hence, having determined mortality rates and *Listerial* burden how these could correlate with haemocyte density was assayed. The objective was to understand larval immunological response to *L. monocytogenes* infections and also determine relative pathogenicity of the isolates. *G. mellonella* infection and haemocyte

quantification was as described above using 75 larvae per bacterial treatment, 30 for controls, and extracting haemolymph from 3 live larvae at each time point for a duration of 7 d. Experiments were carried out three times using different batches of larvae.

Before larval inoculation haemocyte density was quantified, and this represented 0 h for each tested isolate. The results (Fig 6) indicate at 0 h larvae haemocyte density was $3.15 \pm 0.96$ x $10^7$ per ml of haemolymph. In *Listeria* inoculated larvae a gradual decline in haemocyte density was observed within the first 24 h post-infection and thereafter increased to pre-inoculation levels, except in EGD-e and *L. ivanovii*. However, in EGD-e and *L. ivanovii* inoculated larvae the rapid decline continued until 48 h post-infection before stabilising. In contrast, a decline in haemocyte density in PBS-inoculated larvae was only observed in the first 1 h post-infection which gradually increased to pre-inoculation levels within 24 h. Though haemocyte density levels increased in *L. monocytogenes* inoculated larvae after 48 h post-infection this was mostly followed by a gradual decrease and was more apparent in NLmo14 and NLmo20 towards the conclusion of the time course. Nonetheless, mean haemocyte density comparisons amongst treatment groups showed PBS inoculated larvae had higher haemocyte densities than *Listeria* inoculated larvae throughout the course of the investigation, albeit differences were insignificant at most time points. Notwithstanding, though differences in haemocyte density was observed between all strains these were largely between PBS and *L. ivanovii* inoculated larvae at most time points and were more apparent at 24–72 h post-infection. After 7 d incubation however, haemocyte densities amongst treatment groups did not significantly differ from pre-inoculation levels nor were there any significant inter-strain differences at this time point ($p < \geq 0.05$, Two-way ANOVA). Interestingly, while *L. ivanovii* had the least recoverable CFU and exhibited slower growth rates *in vitro* relative to *L. monocytogenes* strains it induced the most sustained decrease in haemocyte density.

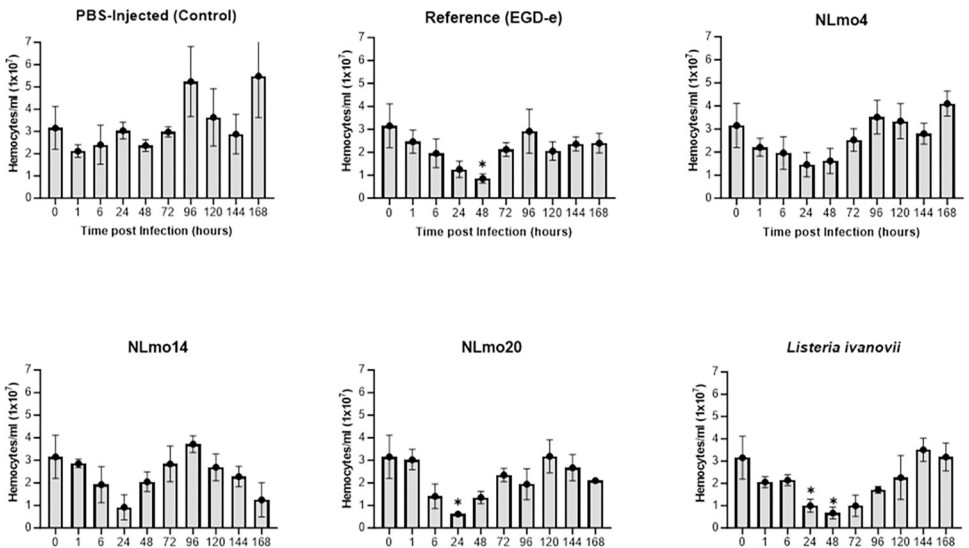

**Fig 6. Fluctuations in haemocyte density in larvae inoculated with Listeria strains.** Bacterial cultures were grown to stationary phase, washed (2x) and resuspended in PBS. Larvae were inoculated with a dose $10^6$ CFU/larvae, and Controls with 20µl of PBS, thereafter, incubated at 37˚C. At each time-point haemolymph was collected from 3 alive larvae, and haemocytes quantified on a haemocytometer chamber. Statistical significance was tested by comparing haemocyte density of Listeria inoculated larvae to PBS inoculated larvae at each time point (* p < 0.05, Two-Way ANOVA). Results represent mean ± SEM of three independent determinations.

## Discussion

Differences in pathogenic potential of *Listeria spp.* is an area of ongoing research and still of limited understanding. More so is our understanding of virulence differences of *L. monocytogenes* strains which, as a consequence, means all *L. monocytogenes* strains are still treated as the same for regulatory purposes. Of the characterised *L. monocytogenes* strains, whose determined virulence have been used to predict the pathogenic potential of other *L. monocytogenes* isolates and disease pathology, nearly all are from clinical sources, whilst isolates from environmental and food sources that could be avirulent or of different virulence profiles to clinical isolates are poorly characterised. This study thus compared the pathogenic potential of *L. monocytogenes* strains (representing lineages I and II) isolated from different stages of the fresh produce supply chain (FPSC) including an isolate from a drainage area in a food processing environment [13] using the *G. mellonella* infection model. Numerous methods have previously been used to determine the virulence potential of *L. monocytogenes* strains, including the chicken embryo test [31–33], Anton's test [34, 35], cell lines based assays [36, 37], laboratory animals [15, 16, 38, 39], and in recent times invertebrate models such as *G. mellonella* [16, 40]. *G. mellonella* is now routinely used as an infection model to assess virulence due to the commonalities it shares with mammalian models, ethical acceptance of its use, ease of handling, cheap to acquire, and low rearing costs [30, 41–43].

Studies using *G. mellonella* as a model host commonly induce infections by inoculating larvae through the haemocoel with viable bacterial strains. Microbial virulence in the model has mostly been evaluated by either comparing the total mortality rates or 50% lethal dose, determining the microbial burden in larvae, larval HISS, or by quantifying haemocyte density and other immunological responses such as expression of AMPs following microbial infections. As each of these variables can be used to determine the pathogenic potential of microorganisms, we recognised that a combination of these assays may enable consistent determination and better discrimination between strains of different virulence profiles. This is also as has been carried out in earlier studies that determined differences in virulence of other bacterial and fungal pathogens, as well as in correlation studies that demonstrated that virulence observed in the model parallels those seen in mammalian infection models [44–47].

Key to our investigation was to test whether the *L. monocytogenes* strains exhibit different virulence potentials in *G. mellonella*. As depicted in Fig 3(A), significant differences in virulence were observed between *L. monocytogenes* strains (p<0.05) based on the rates of larvae mortality induced. *L. monocytogenes* strains induced mortality rates in the range of 65 ± 2.9% to 98.8 ± 1.1% by the end of the 7d time course. All *L. monocytogenes* strains, except NLmo20, were found to cause higher mortality rates in *G. mellonella* larvae than the reference strain used in this study, EGD-e. Also, mortality rates observed for EGD-e corresponded to those reported in other studies at the same time point [25, 48]. A correlation with the *L. ivanovii* strain used in this study as a second reference strain was also found in these studies. These demonstrated the reproducibility and consistency of the *G. mellonella* model but also confirmed that observed differences in virulence between *L. monocytogenes* strains are strain-dependent.

*L. ivanovii* was found to be the least virulent isolate relative to the investigated *L. monocytogenes* strains in this study, causing mean mortality rates of 4.4 ± 4.4% at 7 d post-infection. *L. ivanovii* is known to be of low pathogenic potential relative to *L. monocytogenes* in animal models [48]. This *spp.* of *Listeria* rarely causes clinical cases as of the nine reported clinical cases involving the bacterium since 1970 only one fatality was detailed [49]. The low pathogenicity of *L. ivanovii* is mainly predicated on the low number of virulence factors found in strains of this bacterium relative to *L. monocytogenes* [49]. *L. ivanovii* strains contain only two

of the identified four *Listeria* pathogenic islands (LIPI) with far less virulence factors present in its core genomes as compared to *L. monocytogenes*. Thus, this study in addition to prior *in vivo* studies [25, 48] supported the postulate that virulence of *Listeria* strains is dependent on number of virulence factors found in their genomes. However, the *in silico* predicted virulence of the *L. monocytogenes* strains did not correlate with *in vivo* virulence in our investigation. The number of identified virulence factors from whole genome sequencing (WGS) and other genetic based analysis tools such as PFGE have routinely been used to predict pathogenic potential of *L. monocytogenes* isolates [13, 50, 51]. *L. monocytogenes* isolates used in this study were of different *in silico* predicted virulence stratifications [13]. Of the 42 virulence factors (genes) tested for, NLmo6, NLmo7, NLmo14 and NLmo20 carried the highest number of copies of those [41] whilst NLmo4, NLmo5, and EGD-e (ref strain) possessed the least [31]. Contrary to what we expected, NLmo4 and NLmo5 were the most virulent in *G. mellonella* and the strains of highest predicted virulence mostly exhibited medium (NLmo6 and NLmo14) and low (NLmo20) virulence relative to NLmo4 and NLmo5. Relative to NLmo4 & 5, EGD-e exhibited low virulence similar to that observed with NLmo20. In a literature search no prior studies testing for correlation between *in silico* predictions and *in vivo* virulence for such a wide range of determined virulence factors of *L. monocytogenes* were found. However, clinical strains of different virulence profile for eight *L. monocytogenes* virulence associated genes (*inlA*, *inlB*, and the six genes of LIPI-1) have been evaluated. Click or tap here to enter text. All 27 isolates in that study were equally pathogenic in mice, more so, no correlation was found between *in silico* predictions and strains' virulence in a mouse model [52]. Similar findings have also been reported for *Coxiella burnetii* strains in a mouse model [53]. A *C. burnetti* strain (Guiana Cb175) that had 77 times less virulent genes compared to two other strains, *C. burnetti* German (Z3055) and *C. burnetti* Nine Mile (RSA 493), was found to be the most virulent causing 100% mortality at 4 weeks post-infection whilst at the same time point, a 0% and 75% mortality was reported for the two other strains, respectively. These correlated with the findings of this study, suggesting that the total number of virulence factors is an inaccurate predictor of *in vivo* virulence, at least in *L. monocytogenes*.

*Listeria* strains were investigated for their growth rates *in vitro* and *in vivo* as a means to explain the differences in mortality rates they induce. However, similar growth rates *in vitro* were observed for all strains (S1 Fig) and viable CFU recovered from larvae were only significantly different when *L. ivanovii* was compared to the *L. monocytogenes* strains (Fig 5). Though bacterial burden was not significantly different in larvae inoculated with the most and least virulent *L. monocytogenes* isolates, relative to the mortality rates they induced, the most virulent isolate, NLmo4, had the least recoverable *in vivo* bacterial CFU at 7 d post-infection when compared to the least virulent NLmo20 strain. This indicated that increased numbers of viable bacterial CFU do not correlate with increased mortality rates in the larvae. In addition, *G. mellonella* response to infection and bacterial burden in larvae can also be quantitatively inferred from the rate of larval melanisation [54]. After infection haemocytes are recruited which bind to and limit bacterial growth and dissemination [16, 55]. This opsonisation process that initiates bacterial killing causes nodulations, and an increase in nodule formation is visualised by the increase in larvae melanisation [56, 57]. High bacterial load thus correlates with increased melanisation, which was observed on the dose-dependent assays carried out in this study as well as in reports by other research groups [54, 58]. In the HISS analysis conducted in this study, however, significant differences in larval melanisation between NLmo4 and NLmo20 inoculated larvae was not observed. This further indicates that though mortality rates are dependent on bacterial burden reaching a lethal threshold, mortality rates beyond such a limit are independent of the bacterial replication rates; thus, suggesting an intrinsic expression of

virulence factors by bacterial strains to drive the differences in mortality rates seen between strains of similar growth characteristics.

A correlation between strain virulence and haemocyte density was not observed in *G. mellonella* larvae for the investigated *L. monocytogenes* isolates in this study. It was also recently reported that *Streptococcal* strains of different virulence potentials that included a heat-killed bacterial dose exhibited no significant differences in haemocyte densities [59]. Also, a study using a wild type (WT) *Legionella pneumophila* serogroup I strain (130b) and a mutant (ΔDotA) lacking a type IV secretion system (T4SS) that induced 70% differences in *G. mellonella* mortality rates at 18 h post-infection reported these strains had no significant differences in haemocyte density at the same time point [60]. In contrast, other studies [16, 30, 54] have reported a seemly linear relationship exist between strain virulence and haemocyte density. These suggest mortality rates do not correlate with decreased haemocyte density in all microbial *spp*. The observed differences in this study in relation to the findings by the latter research groups could also be as a result of the methodology used. Whilst we determined total haemocyte counts in this study, live (viable) haemocyte counts were assessed in one of the latter studies [16].

Given the lack of correlation between the high presence of virulence factors and increased larval mortality rates, observed in this study, strains were also evaluated for the presence of non-chromosomal factors from WGS data [13]. The two most virulent strains in this study (NLmo4 and NLmo5) were also the only strains reported to carry plasmid-derived QAC resistant genes [13]. However, these plasmids were not characterised in the study. Nonetheless, plasmids are known reservoirs of bacterial virulence factors and their role in enhancing strain virulence in *in vivo* models, including *G. mellonella*, have been reported in other human pathogens including *Salmonella* [61], *Staphylococcus aureus* [62], *Escherichia coli* [63], and *Pseudomonas aeruginosa* [64]. It was thus theorised that plasmid derived virulence in *L. monocytogenes* strains NLmo4 and NLmo5 could have caused the virulence phenomena observed in this study.

We cannot however rule out that the route chosen to establish *L. monocytogenes* infection could also be an underlying factor to the lack correlation of *in silico* prediction to *in vivo* virulence observed in this study. *L. monocytogenes* is a foodborne pathogen thus infections by the bacterium is primarily via the oral route. In listeriosis when the bacterium is ingested through contaminated food it traverses the intestines into the blood stream, subsequently infecting the liver, cerebrospinal fluid, meninges, and spleen. Failure to clear the infection by immune cells (macrophages, neutrophils, *etc*.) in the liver following this process could lead to severe listeriosis as the bacterium gets released into circulation. However, in this study infections were established by injecting bacterial inoculums directly into the haemocoel which bypasses the required intestinal barrier breach vital for the bacterium to successfully establish an infection. The differences in virulence of *L. monocytogenes* strains could thus be significantly dependent on their ability to traverse the intestinal walls. As such, inoculating bacteria directly into the haemocoel would eliminate a key first-line virulence determiner and minimise the potential of identifying significant differences in virulence that are dependent on the variable virulence factors found in *L. monocytogenes* strains.

However, despite the highlighted limitations the results of the study demonstrate the clear virulence potential of all the *L. monocytogenes* strains tested within the *G. mellonella* model. This supports the current practise of treating all strains of *L. monocytogenes* as a priority pathogen for the food industry which poses significant health risks. The findings also suggest the current United Kingdom public health legislation for food business operators (FBO) of no *L. monocytogenes* in any ready-to-eat foods assessed as able to support the growth of *L.*

*monocytogenes* before the food has left the immediate control of the FBO [65] as appropriate to reduce potential health risks.

## Supporting information

**S1 Fig. Growth of *L. monocytogenes* strains and *L. ivanovii* (absorbance and CFU counts) as a function of time.** BHI broth was seeded with bacterial cultures in stationary phase to a starting absorbance ($OD_{60nm}$) of 0.01 and incubated at 37˚C (200 rpm, aerobic conditions). Absorbance was measured hourly and CFU $ml^{-1}$ at bi-hourly intervals. Results represent individual replicates (CFU $ml^{-1}$) and mean ± SEM values of three independent determinations. (TIF)

## Acknowledgments

We would like to express our sincere gratitude to Edwin R. Moorhouse for providing the bacterial isolates used for this study. The views expressed in this report are those of the authors only.

## Author Contributions

**Conceptualization:** Rosa de Llanos Frutos, Ian Singleton.

**Formal analysis:** Nick Wheelhouse.

**Investigation:** Umaru Bah, Alva Smith, Allen Flockhart.

**Methodology:** Umaru Bah, Alva Smith, Allen Flockhart, Nick Wheelhouse.

**Project administration:** Umaru Bah, Ian Singleton, Nick Wheelhouse.

**Supervision:** Rosa de Llanos Frutos, Samantha Donnellan, Ian Singleton, Nick Wheelhouse.

**Writing – original draft:** Umaru Bah.

**Writing – review & editing:** Umaru Bah, Rosa de Llanos Frutos, Samantha Donnellan, Alva Smith, Allen Flockhart, Ian Singleton, Nick Wheelhouse.

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
