## [Decision Letter · Decision Letter 0]

14 Aug 2024

PONE-D-24-29992The potential virulence of Listeria monocytogenes strains isolated from fresh produce processing facilities as determined by an invertebrate Galleria mellonella model.PLOS ONE

Dear Dr. Wheelhouse,

Thank you for submitting your manuscript to PLOS ONE. After careful consideration, we feel that it has merit but does not fully meet PLOS ONE’s publication criteria as it currently stands. Therefore, we invite you to submit a revised version of the manuscript that addresses the points raised during the review process.

We look forward to receiving your revised manuscript.

Kind regards,

Guadalupe Virginia Nevárez-Moorillón, Ph.D.

Academic Editor

PLOS ONE

Journal Requirements:

"We would like to express our sincere gratitude to Edwin R. Moorhouse for providing the bacterial isolates that were used for this study. We also extend our appreciation to Edinburgh Napier University for funding this study. The views expressed in this report are those of the authors only."

**Additional Editor Comments:**

Please consider the suggestions done by both reviewers

Reviewers' comments:

Reviewer's Responses to Questions

**Comments to the Author**

1. Is the manuscript technically sound, and do the data support the conclusions?

Reviewer #1: Yes

Reviewer #2: Yes

2. Has the statistical analysis been performed appropriately and rigorously? 

Reviewer #1: Yes

Reviewer #2: No

3. Have the authors made all data underlying the findings in their manuscript fully available?

Reviewer #1: Yes

Reviewer #2: Yes

4. Is the manuscript presented in an intelligible fashion and written in standard English?

Reviewer #1: Yes

Reviewer #2: Yes

5. Review Comments to the Author

Reviewer #1: This study aimed to evaluate the virulence of several strains utilizing the Galleria mellonella (G. mellonella) invertebrate infection model, which underscores the importance of delving deeper into the research on Listeria monocytogenes.

(1) When Listeria monocytogenes first appears in the abstract and main text, it should be written in full. Subsequent mentions can utilize the abbreviated genus name.

(2) Regarding the choice of serotype EGD-e, it would be beneficial to clarify the rationale behind this selection. Current research frequently references molecular typing such as ST8, ST87, and ST9, which are known pathogenic types. While the abstract mentions "little is known about the potential virulence of Listeria isolates from the fresh produce environment" and the introduction states "to determine the virulence potential of isolates of the bacterium in order to evaluate the human health risks they pose," these aspects are not extensively discussed in the introduction, especially in relation to molecular typing. Incorporating a discussion on why a serotype-based approach was not taken, given its close association with molecular typing, would strengthen the methodology section.

(3) More detailed information and analysis regarding the HISS index should be included. The basis for its selection and how it contributes to the overall assessment of virulence need to be elaborated upon.

(4) Justification for the inclusion of Listeria ivanovii and further analysis specific to this species should be provided. Explaining its relevance to the study and how it informs our understanding of virulence mechanisms would enrich the discussion.

(5) The discussion should address how existing research has facilitated the development of Dose-Response models and risk assessment studies. Highlighting specific advancements or gaps that this study aims to fill would demonstrate the novelty and significance of your work. Furthermore, discussing the potential implications of your findings for public health policy and risk management strategies would enhance the impact of your research.

Reviewer #2: This paper was based on rigorous academic standards. The introduction provided the necessary background information.

The research methodology for the study is appropriate. However, statistical analyses in material and method section were absent. It should be

6. PLOS authors have the option to publish the peer review history of their article (what does this mean?). If published, this will include your full peer review and any attached files.

Reviewer #1: **Yes: **Qingli Dong

Reviewer #2: No

---

## [Author Response · Author response to Decision Letter 0]

11 Sep 2024

Reviewer #1: This study aimed to evaluate the virulence of several strains utilizing the Galleria mellonella (G. mellonella) invertebrate infection model, which underscores the importance of delving deeper into the research on Listeria monocytogenes.

(1) When Listeria monocytogenes first appears in the abstract and main text, it should be written in full. Subsequent mentions can utilize the abbreviated genus name.

Response: We thank the reviewer and apologise for the oversight. The appropriate changes have been made throughout the document

(2) Regarding the choice of serotype EGD-e, it would be beneficial to clarify the rationale behind this selection. Current research frequently references molecular typing such as ST8, ST87, and ST9, which are known pathogenic types. While the abstract mentions "little is known about the potential virulence of Listeria isolates from the fresh produce environment" and the introduction states "to determine the virulence potential of isolates of the bacterium in order to evaluate the human health risks they pose," these aspects are not extensively discussed in the introduction, especially in relation to molecular typing. Incorporating a discussion on why a serotype-based approach was not taken, given its close association with molecular typing, would strengthen the methodology section.

Response: We thank the reviewer for their comments. In terms of approach we have stated in the introduction that we were investigating the virulence potential of a number of isolates from the food industry which had been previously characterised by molecular typing and WGS (Smith et al, 2019). The study was not to compare methods of typing and in the context of the paper we are not sure that a discussion on a serotype approach would be appropriate. In terms of EGD-e it was used as it is a well characterised and extensively used strain, we have added this clarification (Line 82) citing the relevant reference Becavin et al, 2014. 

(3) More detailed information and analysis regarding the HISS index should be included. The basis for its selection and how it contributes to the overall assessment of virulence need to be elaborated upon.

Response: We have extended the section and added addition appropriate text to highlight the relevance of the HISS (Line 124-132)

(4) Justification for the inclusion of Listeria ivanovii and further analysis specific to this species should be provided. Explaining its relevance to the study and how it informs our understanding of virulence mechanisms would enrich the discussion.

Response: L. invanovii is a related organism that demonstrates significantly lower levels of pathogenicity. We have included a sentence in the methodology (Line 84-87) to clarify this point

(5) The discussion should address how existing research has facilitated the development of Dose-Response models and risk assessment studies. Highlighting specific advancements or gaps that this study aims to fill would demonstrate the novelty and significance of your work. Furthermore, discussing the potential implications of your findings for public health policy and risk management strategies would enhance the impact of your research.

Response: We thank the review for their comments. We have incorporated additional text in the discussion to highlight the relevance of this work to risk management (Lines 482-487)

Reviewer #2: This paper was based on rigorous academic standards. The introduction provided the necessary background information.

The research methodology for the study is appropriate. However, statistical analyses in material and method section were absent. It should be

Response. We thank the reviewer for their comment and have added the appropriate section (Lines 168-174)

---

## [Editor Report · Decision Letter 1]

16 Sep 2024

The potential virulence of Listeria monocytogenes strains isolated from fresh produce processing facilities as determined by an invertebrate Galleria mellonella model.

PONE-D-24-29992R1

Dear Dr. Wheelhouse,

We’re pleased to inform you that your manuscript has been judged scientifically suitable for publication and will be formally accepted for publication once it meets all outstanding technical requirements.

Kind regards,

Guadalupe Virginia Nevárez-Moorillón, Ph.D.

Academic Editor

PLOS ONE
---

## [Editor Report · Acceptance letter]

27 Sep 2024

PONE-D-24-29992R1 

PLOS ONE

Dear Dr. Wheelhouse, 

I'm pleased to inform you that your manuscript has been deemed suitable for publication in PLOS ONE. Congratulations! Your manuscript is now being handed over to our production team.

Kind regards, 

on behalf of

Dr. Guadalupe Virginia Nevárez-Moorillón 

Academic Editor

PLOS ONE